# Effective description of non-equilibrium currents in cold magnetized plasma

Nabil Iqbal[1, *]

[1] *Centre for Particle Theory, Department of Mathematical Sciences,*
*Durham University, South Road, Durham DH1 3LE, UK*

The dynamics of cold strongly magnetized plasma – traditionally the domain of force-free electrodynamics – has recently been reformulated in terms of symmetries and effective field theory, where the degrees of freedom are the momentum and magnetic flux carried by a fluid of cold strings. In physical applications where the electron mass can be neglected one might expect the presence of extra light charged modes – electrons in the lowest Landau level – propagating parallel to the magnetic field lines. We construct an effective description of such electric charges, describing their interaction with plasma degrees of freedom in terms of a new collective mode that can be thought of as a bosonization of the electric charge density along each field line. In this framework QED phenomena such as charged pair production and the axial anomaly are described at the classical level. Formally, our construction corresponds to gauging a particular part of the higher form symmetry associated with magnetic flux conservation. We study some simple applications of our effective theory, showing that the scattering of magnetosonic modes generically creates particles and that the rotating Michel monopole is now surrounded by a cloud of electric charge.

## I. INTRODUCTION

Diverse sets of physical phenomena across vastly different length scales are controlled by the dynamics of magnetic fields in plasma. The description of such plasmas in terms of coarse-grained hydrodynamic degrees of freedom has a long history [1]. A particular regime of a strongly magnetized plasma is obtained when one considers a situation where electric charges are sufficiently plentiful as to screen the electric field to zero, but sufficiently diffuse in that one can ignore their collective stress-energy. Such a regime is conventionally described by the equations of *force-free electrodynamics* (FFE), which describes the non-linear dynamics of magnetic field lines at zero temperature [2–4]. Importantly, this theory has no preferred rest frame. The many applications of this theory include the study of the magnetospheres of compact astrophysical objects [5, 6].

Nevertheless, FFE as conventionally formulated may be considered incomplete. As a coarse-grained theory with no intrinsic scales, it is insufficient to describe by itself astrophysical phenomena such as (e.g.) coherent radiation and the production of particle winds [7–9]. Theoretically, though FFE is clearly an approximation, it is not immediately clear what the small parameter is, nor how exactly it could be improved to better approximate reality.

Thus motivated, and with an eye towards astrophysical applications, [10] (building on a framework constructed in [11]) reformulated force-free electrodynamics as an *effective field theory*. The authors identified a set of symmetry principles and wrote down the most general low-energy action respecting those principles, resulting in a realization of the force-free plasma as a fluid of cold strings. The novel symmetry principle making this possible was that of *generalized global symmetries* [12], and

the utility of such symmetries in the description of magnetized plasma was first articulated in [11]. To leading order in derivatives, the theory of [10] is precisely force-free electrodynamics, where the "strings" are magnetic field lines. There is however an important conceptual difference in that one no longer supposes that the inertia of electric charges is neglected; rather these charges have been "integrated out" in that they no longer appear in the low-energy description. A (slightly) different action principle for an EFT for FFE appears in [13].

Importantly, this effective field theory formalism now allows for the systematic inclusion of higher-derivative corrections to FFE. [11] showed that these corrections can result in qualitatively new physical effects, such as the generation of nontrivial electric fields parallel to the magnetic field, i.e. $\mathbf{E} \cdot \mathbf{B} \neq 0$.

### A. Light charged modes

However, there is reason to believe that the EFT description is still incomplete when applied to some actual physical settings. Imagine the interaction of our FFE plasma with some extra electric charges that are moving ultra-relativistically, perhaps due to their initial conditions, or perhaps because the magnetic field strength is much stronger than the electron rest mass squared, as in magnetars. Under such conditions, it may be a good approximation to consider the electron to be massless.

The massless electrons and positrons will then spin rapidly around the magnetic field lines; quantum mechanically, they will sink into the lowest Landau level. For massless electrons, the energy of the lowest Landau level is exactly zero[1]. Thus the motion of the elec-

---

[*] nabil.iqbal@durham.ac.uk

[1] This occurs through a cancellation between the (negative) energy associated with the Zeeman coupling of the electron spin

tron transverse to the field lines is gapped, but its motion *along* the field lines is gapless[2]. Furthermore, if we work on scales longer than the magnetic cyclotron radius $\ell_B \sim (eB)^{-\frac{1}{2}}$, particles moving along different field lines should be essentially uncorrelated.

These gapless modes are expected to be present at low energies but are clearly not included in the effective description of [10]. One way to understand this is that they are associated with an almost-conserved axial current, which made no appearance in that discussion.

In this work we will construct an effective description of such light electric charges and their interaction with the plasma. We are motivated primarily by practical considerations, and so take the viewpoint that these are simply "extra" degrees of freedom that are not in equilibrium with FFE plasma: thus we will sometimes refer to them as "non-equilibrium" charges. As they move freely only along the two-dimensional world-sheet swept out by the field line in spacetime, a great deal of intuition for this problem can be obtained from the *bosonization* of two-dimensional fermions, where a bosonic collective mode captures the dynamics of the fermionic charge density.

We similarly introduce a new effective 4d bosonic field $\Theta(x)$ that is essentially a bosonized version of the electric charge current $j^{\text{el}}$ along each field line.

$$j_{\text{el}}^{\sigma} = -\frac{1}{2} n_{\mu\nu} \epsilon^{\sigma\rho\mu\nu} \partial_\rho \Theta(x) \tag{1}$$

where here $n_{\mu\nu}$ is a unit-norm tensor introduced in [10] that is proportional to the electromagnetic field strength $F_{\mu\nu}$. We will couple this field in a universal manner to the FFE degrees of freedom. A key technical point is the identification of a new symmetry principle that ties $\Theta$ to the low-energy degrees of freedom of [10] in a way that confines the charges to move along field lines.

Though our implementation in terms of EFT is novel, similar ideas have appeared before. In particular, building on work in [15, 16], [17] recently performed a microscopic construction of a similar collective field by directly bosonizing the Landau levels of a 4d Dirac fermion along a homogenous magnetic field. However, that work was a perturbative computation, and the existence of a force-free limit and the validity of various approximations is not clear to us. Our hydrodynamic approach is an attempt to directly arrive at an effective low-energy description.

––––––––

to the magnetic field and the (positive) energy of the zero-point cyclotron motion about the field line. This cancellation is required by index theorems that govern the realization of the 4d axial anomaly.

[2] At zero EM coupling this statement is obvious. At weak EM coupling, the situation is somewhat subtle: a single one of these zero modes – the "center of mass", i.e. the appropriately defined uniform sum over states in the lowest Landau level – acquires a mass by a coupling to the zero mode of the photon (see e.g. [14]) but the majority of them remain massless.

## B. Summary

We now summarize the remainder of the paper. In Section II we review some useful background material. In Section III we present our effective description of electric charges. In Section IV we specialize to a particular theory (fixing various thermodynamic functions that appear in the general framework) and in Section V we present some simple applications, i.e. we demonstrate that the scattering of magnetosonic modes results in particle creation and that the Michel monopole is now surrounded by a halo of non-equilibrated charges. In Section VI we conclude with some directions for future research.

## II. BACKGROUND

Here we review some background material. The reader who is completely familiar both with conventional 2d bosonization and with the EFT construction of FFE in [10] should skip to Section III, where the addition of gapless electric charges is presented.

### A. Review of 2d bosonization

We briefly discuss 2d bosonization. This is textbook material, see e.g. [18]. Recall the Schwinger model, i.e. a massless 2d Dirac fermion coupled to 2d electromagnetism:

$$S_{2d} = \int d^2x \left( \bar{\psi} \left( i\slashed{\partial} - i\slashed{A} \right) \psi - \frac{1}{4e^2} F^2 \right) \tag{2}$$

This theory has a vector and axial current, written as

$$j_V^\mu = \bar{\psi}\gamma^\mu\psi \qquad j_A^\mu = \bar{\psi}\gamma^\mu\gamma^3\psi \tag{3}$$

The vector current is coupled to the gauge field $A_\mu$, and is what we conventionally call "electric charge". The axial current is not actually conserved; a one-loop computation shows that it is is afflicted by the well-known 2d axial anomaly:

$$\partial_\mu j_A^\mu = \frac{1}{2\pi} \epsilon^{\mu\nu} F_{\mu\nu} . \tag{4}$$

We can precisely reformulate this fermionic system as a theory of a single boson $\phi$:

$$S_{2d} = \int d^2x \left( -\frac{1}{8\pi}(\partial\phi)^2 - \frac{1}{4e^2}F^2 + \frac{1}{2\pi}A_\mu\epsilon^{\mu\nu}\partial_\nu\phi \right) \tag{5}$$

In this formulation, the vector and axial currents are

$$j_V^\mu = \frac{1}{2\pi}\epsilon^{\mu\nu}\partial_\nu\phi \qquad j_A^\mu = \frac{1}{4\pi}\partial^\mu\phi . \tag{6}$$

Note that the bosonic field $\phi$ provides an alternative description of the dynamics of the charge sector, which may

be more convenient for certain purposes. For example, the axial anomaly (4) is now visible at the classical level, as it is equivalent to the equation of motion of the field $\phi$. Note also that the vector electric current is *identically* conserved, unlike in the fermionic description. Many features of this story will reappear in our construction below.

## B. Review of EFT of FFE

We now turn to FFE, which is often presented as a theory of the Maxwell field strength tensor $F^{\mu\nu}$, supplemented with the condition that bare electric charges are present in sufficient quantities to screen the electric field, but in insufficient quantities for their energy-momentum exchange with the electromagnetic fields to matter (see e.g. [4]).

In [10], a different effective theory viewpoint on FFE was presented. We review this briefly here, assuming that the reader is already somewhat familiar with that work. As usual in EFT, we begin by identifying conserved quantities. One of them is the usual stress tensor $T^{\mu\nu}$, whose conservation follows from general covariance in the presence of a background metric $g_{\mu\nu}$. More interestingly, for a theory including dynamical magnetic fields, the magnetic flux $J^{\mu\nu} \equiv \frac{1}{2}\epsilon^{\mu\nu\rho\sigma}F_{\rho\sigma}$ is also a conserved quantity, as the Bianchi identity guarantees that we have

$$\nabla_\mu J^{\mu\nu} = 0 \tag{7}$$

We note that the symmetry associated with this conserved quantity may be somewhat unfamiliar and is called a *generalized global symmetry* [12]. Such symmetries are present whenever one has a conserved density of extended objects (such as magnetic field lines); they were initially understood in the context of non-Abelian gauge theory, and have recently begun to be used to constrain hydrodynamic theories [11, 13, 19–27].

The main idea of [10] is to realize this symmetry *not* on the microscopic photon and electrons, but rather on a different set of low-energy collective fields, which are taken to be two scalar fields $\Phi_{1,2}$ and a vector field $a_\mu$ called the worldsheet magnetic photon. The simultaneous level sets of $\Phi_{1,2}$ determine the magnetic field worldsheets, and $da$ measures the magnetic flux on each worldsheet.

It is very useful to couple $J^{\mu\nu}$ to an external *fixed* source field $b_{\mu\nu}$. The symmetry associated with $J$ is then implemented by demanding that the theory is invariant under the following simultaneous transformation of dynamical field $a$ and source $b$:

$$b \to b + d\lambda \qquad a \to a + \lambda \tag{8}$$

where $\lambda$ is an arbitrary 1-form[3].

It will be very important for our later purposes that this external source $b$ may be interpreted as an external electric current density via

$$j_{\text{el}}^\sigma = -\frac{1}{2}\epsilon^{\sigma\rho\mu\nu}\partial_\rho b_{\mu\nu}. \tag{9}$$

This relation is explained in detail in [11], and arises physically from the idea that the natural source for a theory of dynamical electromagnetism is a fixed external charge density. Note that the current is identically conserved.

Given the external sources $b$ and $g$, we compute the conserved quantities from the action as

$$T^{\mu\nu} \equiv \frac{2}{\sqrt{-g}}\frac{\delta S}{\delta g_{\mu\nu}} \qquad J^{\mu\nu} \equiv \frac{2}{\sqrt{-g}}\frac{\delta S}{\delta b_{\mu\nu}}. \tag{10}$$

In addition to this microscopic symmetry, we also demand that the theory is invariant under the following *emergent* symmetries of the description:

- Field sheet relabelings, i.e reparametriztions

$$\Phi_I \to \Phi_I'(\Phi_I) \tag{11}$$

- String-dependent 1-form shifts:

$$a \to a + \omega(\Phi_1, 2) \tag{12}$$

Here $\omega$ is an arbitrary function of $\Phi_{1,2}$, and so varies from worldsheet to worldsheet. These symmetries should be thought of as characterizing the low-energy phase that is force-free electrodynamics.

We now write down the most general effective action that is invariant under these symmetries. It is convenient to introduce the following 2-form and its magnitude.

$$S_{\mu\nu} \equiv \nabla_{[\mu}\Phi_1\nabla_{\nu]}\Phi_2 \qquad s = \sqrt{\frac{S_{\mu\nu}S^{\mu\nu}}{2}} \tag{13}$$

We further construct the binormal $n$ and the volume form $\varepsilon$ on the foliation

$$n_{\mu\nu} = \frac{S_{\mu\nu}}{s} \qquad \varepsilon_{\mu\nu} = \frac{1}{2}\epsilon_{\mu\nu\rho\sigma}n^{\rho\sigma} \tag{14}$$

which satisfy

$$n_{\mu\nu}n^{\mu\nu} = -\varepsilon_{\mu\nu}\varepsilon^{\mu\nu} = 2, \qquad n \wedge n = \varepsilon \wedge \varepsilon = 0 \tag{15}$$

$S_{\mu\nu}$ is invariant only under volume-preserving reparametrizations of the $\phi_I$, but both $n$ and $\varepsilon$ are invariant (up to a sign) under all reparametrizations.

---

[3] The nonlinear transformation of $a$ is similar to that of a 1-form

Goldstone mode [28, 29] but the 1-form shift (12) means that the symmetry is not spontaneously broken in this phase: this is a generalization of a similar "chemical-shift" symmetry which plays the same role in effective actions for conventional hydrodynamics [30].

It will often be convenient to work with the projectors parallel and perpendicular to the foliation:

$$h_{\mu\nu} = -\varepsilon_{\mu\rho}\varepsilon_\nu{}^\rho, \qquad h_{\mu\nu}^\perp = n_{\mu\rho}n_\nu{}^\rho. \tag{16}$$

The only invariant scalar to lowest order in derivatives is

$$\mu = \frac{1}{2}\varepsilon^{\mu\nu}\left(b_{\mu\nu} - \partial_\mu a_\nu + \partial_\nu a_\mu\right). \tag{17}$$

We note that the microscopic symmetry requires that $a$ always appear in the combination $b - da$, and the string-dependent shift further requires that this be projected against $\varepsilon$.

We may now use this scalar to write down an invariant action; for example, the leading order action is

$$S_0[\Phi, a; g, b] = \int d^4x \sqrt{-g}\, p(\mu). \tag{18}$$

where $p$ is an arbitrary function of $\mu$. Borrowing terminology from hydrodynamics, we call this the "ideal" action: from here we can use (10) to construct the ideal stress tensor and magnetic flux as

$$J_0^{\mu\nu} = \rho\varepsilon^{\mu\nu} \qquad T_0^{\mu\nu} = pg^{\mu\nu} - \mu\rho h^{\mu\nu}, \tag{19}$$

where $\rho = \frac{dp}{d\mu}$. The equations of motion for this theory are simply the conservation equations for $T$ and $J$, which are:

$$\nabla_\mu T^{\mu\nu} = \frac{1}{2}(db)^\nu{}_{\rho\sigma}J^{\rho\sigma}, \qquad \nabla_\mu J^{\mu\nu} = 0, \tag{20}$$

If we further make the choice $p(\mu) = \frac{1}{2}\mu^2$, this theory is precisely equivalent to usual FFE. Recall that $\mathbf{E}\cdot\mathbf{B} = \frac{1}{8}\epsilon_{\mu\nu\rho\sigma}J^{\mu\nu}J^{\rho\sigma}$; if we take $J$ to be given by its ideal expression (19), we see that $\mathbf{E}\cdot\mathbf{B} = 0$, as required for ideal FFE.

Of course, the value of the EFT formalism is that it is now possible to systematically include corrections to FFE, simply by adding higher derivative terms to the action (18). We emphasize one particular class of corrections, e.g. consider the following invariant second order term:

$$S_R[\Phi, a; g, b] = \frac{1}{2}\int d^4x \sqrt{-g}\, R(\mu)\nabla^\alpha \epsilon^{\beta\gamma}(db)_{\alpha\beta\gamma} \tag{21}$$

This results in the following correction $J_R$ to the ideal flux (19):

$$J = J_0 + J_R \qquad J_R^{\mu\nu} = -3\nabla_\sigma\left(R(\mu)\nabla^{[\sigma}\epsilon^{\mu\nu]}\right) \tag{22}$$

In the presence of this correction, we generically have that $\mathbf{E}\cdot\mathbf{B} \neq 0$; thus it creates an accelerating electric field. However, this correction to $J$ is identically conserved, meaning that within this theory it merely alters the relationship between the flux $J$ and the dynamical fields without changing the equations of motion. As one might expect, this will change once we add light electric charges for this electric field to pull on in the next section.

From now on, we will refer to the theory reviewed here as the "FFE sector", with total action (including possible higher-derivative corrections) $S_{\mathrm{FFE}}[\Phi, a; g, b]$.

## III. EFFECTIVE THEORY OF NON-EQUILIBRIUM CURRENTS

### A. Electric charge and symmetries

We now finally turn to the addition of non-equilibrium massless charges; as motivated above, we would like to couple the FFE EFT described above to a density of massless electric charges which are confined to move along field lines. Note that the key object needed to perform this coupling already exists within the FFE EFT; in particular, through (9), we know that the external source $b$ already has the interpretation as an electric charge density.

We thus introduce a new *dynamical* field $\Theta(x)$, and couple it to the FFE degrees of freedom through $b$. We write

$$b(\Theta) = \bar{b} + \Theta(x)n \tag{23}$$

where $\bar{b}$ is now the fixed external source, and where $n$ is the binormal defined in (14). To get some intuition for this choice, take $n$ to be constant (corresponding to a constant magnetic field): from (9) we see that the electric current arising from $\Theta$ is

$$j_{\mathrm{el}}^\sigma = -\frac{1}{2}n_{\mu\nu}\epsilon^{\sigma\rho\mu\nu}\partial_\rho\Theta(x) \tag{24}$$

As desired, this is exactly a current propagating along the magnetic field directions. Indeed comparing it to the expression for the 2d vector current (6), we see that $\Theta$ plays a role very similar to the field $\phi$ appearing in the bosonized description of the 2d Dirac fermion.

It is however not enough to demand that the current flow mostly along the worldsheet. As motivated earlier, the current on each field-sheet should be also essentially uncorrelated, as we are studying infrared physics on scales much longer than the magnetic cyclotron length. To ensure that derivatives perpendicular to the worldsheet do not enter the theory, we further demand invariance under the following symmetry:

- *String-dependent scalar shift:*

$$\Theta(x) \to \Theta(x) + s(x)f(\Phi_1, \Phi_2) \tag{25}$$

Here $f$ is an arbitrary function of $\Phi_{1,2}$, whereas $s(x)$ was defined in (13). As we will discuss, this symmetry is closely related to axial charge conservation. We also discuss the rationale for the factor of $s(x)$ below.

We now write the full action of the system as

$$S = S_\Theta[\Phi, a, \Theta; g, b(\Theta)] + S_{\mathrm{FFE}}[\Phi, a; g, b(\Theta)] \tag{26}$$

where the notation $b(\Theta)$ serves to stress that everywhere $b$ is written in terms of $\Theta$ as in (23). Here $S_\Theta$ is a new term that describes the dynamics of $\Theta$ itself, whereas $S_{\mathrm{FFE}}$ is the FFE theory constructed previously. Note that $S_{\mathrm{FFE}}$ depends on $\Theta$ only through $b(\Theta)$. This is a

kind of "minimal coupling", indicating that the charge degrees of freedom affect the FFE sector only through their electric charge density (9).

We now discuss the realization of the symmetries. In particular, it is not at all clear that the FFE sector together with the coupling (23) is itself invariant under the shift symmetry (25). To see that it is, note that under (25), $b$ shifts as

$$b \to b + f(\Phi_1, \Phi_2) d\Phi_1 \wedge d\Phi_2 \qquad (27)$$

where we have used that $sn = d\Phi_1 \wedge d\Phi_2$.

Now the magnetic shift symmetry (8) guarantees that $b$ can enter the action only as $db$, or together with the magnetic worldsheet photon $a$ in the combination $(b - da)$. $db$ is manifestly invariant under (27). Furthermore, the 1-form string-dependent shift of $a$ (12) means that $b - da$ must be projected down onto the worldsheet (e.g. as in (17)). However this projection is also invariant under (27), as $d\Phi_{1,2}$ is perpendicular to the worldsheet.

The upshot is that any FFE theory where $b$ enjoys the symmetries recorded in the previous section can be coupled to a $\Theta$ field as in (23) and is *automatically* invariant under (25). The factor of $s$ present in that expression is crucial for this invariance. One way to understand this is that $f(\Phi_1, \Phi_2)$ is not a scalar in the space of $\Phi_{1,2}$ but rather a 2-form, and $s$ transforms in the appropriate manner to allow us to add it to a true scalar $\Theta$ in (25).

## B. Charge dynamics

Having coupled $\Theta$ to the magnetic field, we now turn to the dynamics of $\Theta$ itself. We would like to construct a kinetic term for $\Theta$. Arbitrary derivatives of $\Theta$ are clearly forbidden by the shift symmetry (25). The 3-form $d(\Theta n)$ is however invariant under all symmetries, and we can use it to construct a kinetic term. At leading order in derivatives, the only candidate is

$$S_\Theta = -\frac{1}{4} \int d^4 x \sqrt{-g} Q(\mu) \nabla_{[\mu} \left( n_{\rho\sigma]} \Theta \right) \nabla^{[\mu} \left( n^{\rho\sigma]} \Theta \right) \qquad (28)$$

Here $Q(\mu)$ is an arbitrary function of $\mu$. Again to obtain intuition consider the case where $n$ is constant, in which case we find

$$S_\Theta = -\frac{1}{2} \int d^4 x \sqrt{-g} Q(\mu) h^{\mu\nu} \nabla_\mu \Theta \nabla_\nu \Theta \qquad (29)$$

where from (16) $h_{\mu\nu}$ is a projector parallel to the worldsheet; thus in equilibrium $\Theta$ becomes effectively a collection of two-dimensional fields, each with dynamics only on the worldsheet.

We note that in situations where $n$ is not constant, then both derivatives and electric current off the worldsheet will appear; however the precise manner in which this happens is dictated by the symmetry principles above, and is thus a prediction of our EFT.

We now discuss the equations of motion, starting with that for $\Theta$. The variation of the total action with respect to $\Theta$ takes the form

$$\delta_\Theta S = \delta_\Theta S_\Theta + \frac{\delta S}{\delta \bar{b}_{\mu\nu}} n^{\mu\nu} \qquad (30)$$

where the second term arises from the implicit dependence of $b$ on $\Theta$ in (23). Putting in the explicit form of $S_\Theta$ and using the definition of $J$ in (35), we find the following equation of motion:

$$\nabla_\alpha \left[ \nabla^{[\alpha} \left( n^{\beta\sigma]} \Theta \right) Q(\mu) \right] n_{\beta\sigma} + J^{\mu\nu} n_{\mu\nu} = 0 \qquad (31)$$

This is a wave equation $\Theta$ on the field-sheets, sourced by a term that depends on the magnetic field. To understand the source term, note that within our construction, magnetic domination means that $J$ always points mostly in the direction of $\varepsilon$ – indeed at ideal order, we see from (19) that it is precisely proportional to $\varepsilon$. As $n = \star\varepsilon$, the source term $J^{\mu\nu} n_{\mu\nu}$ is essentially proportional to $J \wedge J \sim \mathbf{E} \cdot \mathbf{B}$, i.e. to the presence of unscreened accelerating electric fields. As we describe, under some circumstances this sourced wave equation can be interpreted as pair creation.

## C. Axial anomaly and pair creation

This equation of motion is closely related to the shift symmetry (25). Recall that this is actually infinitely many symmetries, parametrized by a free function $f(\Phi_1, \Phi_2)$; thus this leads to infinitely many Noether charges, one on each worldsheet. As we will argue, this can be thought of as an independent *axial* current on each field line.

To understand this, it is instructive to consider the contribution to this Noether current not from the full action, but only from $S_\Theta$; denoting this by $j_f^\mu$, we find

$$j_f^\alpha = \nabla^{[\alpha} \left( n^{\beta\sigma]} \Theta \right) Q(\mu) n_{\beta\sigma} f(\Phi_1, \Phi_2) s(x) \qquad (32)$$

As we have neglected the contribution from the FFE sector, this is not quite conserved, and instead we have

$$\nabla_\alpha j_f^\alpha = -J^{\mu\nu} n_{\mu\nu} f(\Phi_1, \Phi_2) s(x) \qquad (33)$$

In the case $f = 1$, this is equivalent after some manipulation to the usual $\Theta$ equation of motion (31), though we can only write the left-hand side of that equation as a divergence if we re-introduce the field $s(x)$.

This non-conservation equation may be understood as a hydrodynamic manifestation of the following *microscopic* Adler-Bell-Jackiw anomaly equation arising in the theory of fermions coupled to QED:

$$\nabla_\mu j_A^\mu = -\frac{1}{16\pi^2} \epsilon^{\mu\nu\rho\sigma} F_{\mu\nu} F_{\rho\sigma} . \qquad (34)$$

where $j_A^\mu$ is the axial current density $\bar{\psi}\gamma^\mu\gamma^5\psi$.

We see that the role of $j_A^\mu$ is played approximately by $\partial_\mu \Theta$, weighted by $Q(\mu)$ and projected in an appropriate manner onto the worldsheet. Thus $\Theta$ is something like a worldsheet axion [31].

It is instructive to examine how this equation relates to pair creation in a magnetic field; our discussion here is closely related to the chiral magnetic effect [32] (see [33] for a review). Consider pair creation of a massless $e^\pm$ pair by a nonzero $\mathbf{E} \cdot \mathbf{B}$. Energetically, the electron and positron will want to appear in their lowest Landau level. The spin in this lowest Landau level is correlated with the magnetic field: for the electron it is aligned and for the positron it is anti-aligned. Under the influence of the electric field, the electron and positron will move off in opposite directions along the magnetic field; thus for both particle and anti-particle the spin is correlated with the motion. They both have the same helicity, and the whole process thus results in the creation of net *axial* charge. [4] Microscopically, this process is governed by the ABJ anomaly (34), which directly relates $\mathbf{E} \cdot \mathbf{B}$ to axial charge creation.

In our hydrodynamic setup, we are describing the same process, except at scales much longer than the magnetic cyclotron length. Thus each field line is independent, and no charge can move from one field line to the next, so we obtain the stronger relation (33), corresponding to an independent (non) conservation law on each field line. It is extremely interesting to see the physics of the anomaly emerging naturally from our purely hydrodynamic construction. We note also that the resulting expression is very similar to the bosonization of the 2d axial anomaly discussed around (6), except that the structure induced by $\Phi_{1,2}$ provides the information required to sew different field sheets together.

Finally, while the qualitative physics of the anomaly does appear from our construction, the right-hand side of the equation is not *precisely $F \wedge F$*, but is only proportional to it, where the constant of proportionality is "dynamical" in that it depends on details of the thermodynamic function $p(\mu)$. There does not appear to be a protected anomaly coefficient. This is related to the fact that the right hand side of the anomaly equation (34) is a dynamical operator and not an external source (as it is in usual examples of anomalous hydrodynamics [34, 35]), and thus in a theory of dynamical electromagnetism axial current is simply genuinely non-conserved. Similar issues related to non-universality have been studied in [36].

---

[4] To be more precise: in standard conventions, the particle and anti-particle both have the same helicity, but for an anti-particle the chirality is defined to be the opposite of the helicity [32]. The left-hand side of the (integrated) anomaly equation may be understood as (the sum of the numbers of particles and anti-particle with right-handed helicity) minus (the sum of the numbers of particles and anti-particles with left-handed helicity).

## D. Effect of collective mode on FFE

Having exhaustively discussed the equation of motion of $\Theta$, we return finally to the FFE sector. The stress tensor and magnetic flux of the system are now obtained by differentiation of the total action with respect to $g$ and $\bar{b}$ respectively:

$$T^{\mu\nu} \equiv \frac{2}{\sqrt{-g}} \frac{\delta S}{\delta g_{\mu\nu}} \qquad J^{\mu\nu} \equiv \frac{2}{\sqrt{-g}} \frac{\delta S}{\delta \bar{b}_{\mu\nu}}. \qquad (35)$$

Note that as $\mu$ depends on $\bar{b}$, both $J$ and $T$ receive contributions from $S_\Theta$.

As before, the the equations of motion are simply the conservation equations for the currents:

$$\nabla_\mu T^{\mu\nu} = \frac{1}{2}(d\bar{b})^\nu{}_{\rho\sigma} J^{\rho\sigma}, \qquad \nabla_\mu J^{\mu\nu} = 0, \qquad (36)$$

with the modification that generically both $T$ and $J$ will now receive extra contributions from the $\Theta$ sector. We will compute these contributions for a particular choice of theory below. Note that it is the external source $\bar{b}$ that appears on the right hand side of the non-conservation equation for $T$; in most situations it is set to 0.

Finally, we discuss a general feature of the equations of motion. Let us imagine that the FFE sector has no higher derivative corrections and is given by (18). In that case $J \propto \epsilon$, and thus the source term in (31) is zero. It is then consistent to set $\Theta$ to 0, and thus all solutions to FFE remain solutions of the coupled theory. Linearized fluctuations of $\Theta$ about any FFE solution will decouple.

On the other hand, in the presence of higher derivative corrections such as (21), $\mathbf{E} \cdot \mathbf{B}$ is no longer zero, and now the source term in (31) will turn on $\Theta$. As expected, accelerating electric fields can have a dramatic effect on free electric charges, including the hydrodynamic manifestation of the pair creation process discussed previously. We will study some aspects of this below.

## E. Formal aspects

We now note some formal aspects of the above construction. In particular, the sufficiently universally-minded reader may be somewhat puzzled that after going through all the effort of constructing an effective symmetry-based description of FFE in [10], we then perform a brutal operation – i.e. couple in "extra" massless charges – that is motivated not by global symmetry structure but rather mostly by phenomenological considerations.

We do not really feel that we have a completely satisfactory response to such a reader, but we note that at a formal level this construction corresponds to *gauging* a particular part of the 1-form symmetry associated with magnetic flux conservation.

To be more precise, in the construction of [10], an external $b$ field couples to the 2-form current $J$. By making

part of this $b$ field dynamical as in (23),

$$b = \bar{b} + \Theta(x)n \qquad (37)$$

and providing it with a kinetic term, we are gauging "the part of the 2-form current perpendicular to the magnetic field lines." This means that excitations of those gauged components of $J$ – i.e. precisely those that create $\mathbf{E} \cdot \mathbf{B}$ – are now parts of a gauge current and not a global one, and $\Theta$ is a new sort of gauge field for these components of $J$.

As usual in gauge theory, exciting gauge charges costs energy in terms of the gradients of $\Theta$, i.e. through the equation of motion (31), which can be thought of as Gauss's law for the new gauge symmetry. (One can compare this to the solution in conventional weak-coupling electrodynamics, where excitations of the "gauged" electric charge cost energy in terms of the gradients of the vector potential).

We find this somewhat suggestive but still incomplete, as the identification of which part of the 1-form symmetry we gauge is made through the binormal $n$, which is itself still a low-energy construct. Thus we do not see a purely universal way to characterize this gauging procedure. It would be extremely interesting if one could be found and related to the structure of the axial anomaly discussed in the previous subsection.

## IV. SPECIFIC THEORY

We briefly summarize. Given an EFT construction of FFE governed by an effective action $S_{\text{FFE}}$, there is a way to "minimally" couple it to free electric charges confined to move along string worldsheets, where the dynamics of these charges is given by $S_\Theta$, as in (26):

$$S = S_\Theta[\Phi, a, \Theta; g, b(\Theta)] + S_{\text{FFE}}[\Phi, a; g, b(\Theta)] . \qquad (38)$$

For concreteness, in the remainder of this paper we will work with the specific theory given by (38), where the FFE sector is given by the choice

$$S_{\text{FFE}} = S_0 + S_R \qquad (39)$$

with $S_0$ and $S_R$ given by (18) and (21) respectively. $S_0$ and $S_\Theta$ are the unique choices at leading order in derivatives. The choice of $S_R$ (and no other higher derivative corrections) is arbitrary, and was made purely for convenience to illustrate the physics of pair creation; we do not expect the qualitative physics to change if generic higher derivative corrections are added.

We now discuss length scales. This theory contains three arbitrary functions of $\mu$: $p(\mu), Q(\mu)$, and $R(\mu)$. Using the fact that both $\mu$ and $\Theta$ have mass dimension 2 (for the latter, see (23)) the leading order expansion of each of these quantities in powers of $\mu$ is:

$$p(\mu) = \frac{1}{2}\mu^2 + \cdots \qquad Q(\mu) = \frac{Q_0}{|\mu|} + \cdots \qquad R(\mu) = \frac{\mu}{\Lambda^2} + \cdots \qquad (40)$$

where $\Lambda$ is some UV mass scale. In all concrete computations from here on, we will restrict to just the leading term in each of these expressions.

Notably, both $p$ and $Q$ are scale-free to leading order. The non-analytic behavior of $Q(\mu)$ as a function of $\mu$ may seem surprising. To understand this, note that in the FFE limit the magnitude of the magnetic field $B = \mu$. Microscopically, the physics of Landau levels is indeed non-analytic as a function of the magnetic field; for example, the density of states of Landau zero modes is controlled by $e|B|$, and this is the ultimate origin of the non-analyticity in $\mu$.

At the level of the EFT, $Q_0$ is a free parameter controlling the strength of interactions of $\Theta$ with the FFE degrees of freedom. In principle it can be determined from a UV description; in Appendix A we discuss a preliminary attempt at matching with the microscopic treatment of [17], where the UV completion is provided by Dirac electrons coupled to QED with electromagnetic coupling $e$. In that case we find

$$Q_0 = \frac{2\pi^2}{e^3} \qquad (41)$$

This identification comes with caveats, and we refer the interested reader to the Appendix for a full discussion. We will keep $Q_0$ arbitrary in what follows.

Thus this minimal theory has a single length scale $\Lambda^{-1}$; it controls the magnitude of possible $\mathbf{E} \cdot \mathbf{B}$, and thus can be interpreted as defining the scale over which the perfect screening of $E$ in the plasma breaks down. An unscreened $E$ will pair-create charges (parametrized by $\Theta$) through the source term in (31). These charges will then move about, interacting nonlinearly with the plasma in a way that is captured by the evolution equations below.

We will provide an extremely preliminary discussion of the possible physics arising from this in some applications to simple geometries (wave scattering about a homogenous background, and the rotating Michel monopole) below.

### A. Detailed expressions for stress tensor and flux

For completeness, we write down the stress tensor and flux. We find

$$J^{\mu\nu} = (\rho + \rho_\Theta)\varepsilon^{\mu\nu} - 3\nabla_\sigma \left( R(\mu)\nabla^{[\sigma}\varepsilon^{\mu\nu]} \right) \qquad (42)$$

where as before $\rho = \frac{dp}{d\mu}$, but $\rho_\Theta$ is a scalar correction at $\mathcal{O}(\Theta^2)$ to the effective magnetic flux arising from the $\mu$-dependence of the $\Theta$ kinetic term:

$$\rho_\Theta \equiv -\frac{1}{4}\frac{dQ}{d\mu}\nabla_{[\mu}\left(n_{\rho\sigma]}\Theta\right)\nabla^{[\mu}\left(n^{\rho\sigma]}\Theta\right) \qquad (43)$$

As the expression for $T^{\mu\nu}$ is somewhat lengthier, we break it into several pieces:

$$T^{\mu\nu} = T_0^{\mu\nu} + T_R^{\mu\nu} + T_\Theta^{\mu\nu} \qquad (44)$$

where $T_0^{\mu\nu}$ was given in (19) and where $T_R$ and $T_\Theta$ arise from differentiating (21) and (28) with respect to $g$ respectively, and are:

$$T_{\mu\nu}^R = \frac{1}{2}\left(-R(\mu)g_{\mu\nu} + \mu\frac{dR}{d\mu}h_{\mu\nu}\right)\nabla^\alpha\epsilon^{\beta\gamma}(db)_{\alpha\beta\gamma} + \\ 3R(\mu)\nabla_{[\mu}\epsilon_{\rho\sigma]}(db)_{\nu\alpha\beta}g^{\rho\alpha}g^{\sigma\beta} \tag{45}$$

where we remind the reader that $b = \bar{b} + \Theta n$ with $\bar{b}$ the fixed external source, and

$$T_{\mu\nu}^\Theta = -\frac{1}{4}\Bigg[\left(-Q(\mu)g_{\mu\nu} + \mu\frac{dQ}{d\mu}h_{\mu\nu}\right)\frac{(\nabla n\Theta)^2}{2} + \\ 3Q(\mu)\nabla_{[\mu}\left(n_{\rho\sigma]}\Theta\right)\nabla_{[\nu}\left(n_{\alpha\beta]}\Theta\right)g^{\rho\alpha}g^{\sigma\beta}\Bigg] \tag{46}$$

where the notation $(\nabla n\Theta)^2 \equiv \nabla_{[\mu}\left(n_{\rho\sigma]}\Theta\right)\nabla^{[\mu}\left(n^{\rho\sigma]}\Theta\right)$.

To simplify the above computations, it was helpful to note that the variation of $n$ with respect to $g$ is:

$$\delta_g n_{\mu\nu} = -\frac{1}{2}n_{\mu\nu}h_{\alpha\beta}^\perp\delta g^{\alpha\beta} \tag{47}$$

This means that the metric variation of the combination $\Theta n$ is proportional to $n$, and thus takes the same form as a variation of $\Theta n$ with respect to $\Theta$. Such variations vanish on-shell, i.e. when the $\Theta$ equations of motion (31) are imposed. In this situation it is then consistent to set *all* metric variations of $(\Theta n)$ to zero when computing the stress tensor. We have done so in the above expressions, which are thus valid only on-shell. (Generically they contain other terms proportional to the $\Theta$ equation of motion multiplying $h_\perp$).

To recap: the degrees of freedom can be taken to be the FFE variables $\epsilon, \mu$ together with the charge mode $\Theta$. The equations of motion are the conservation equations (48), which for convenience we reproduce in the case where $\bar{b} = 0$:

$$\nabla_\mu T^{\mu\nu} = 0, \qquad \nabla_\mu J^{\mu\nu} = 0, \tag{48}$$

together with the $\Theta$ equation of motion, which we write out explicitly for arbitrary $R(\mu)$.

$$\nabla_\alpha\left[\nabla^{[\alpha}\left(n^{\beta\sigma]}\Theta\right)Q(\mu)\right]n_{\beta\sigma} - 3\nabla_\sigma\left(R(\mu)\nabla^{[\sigma}\varepsilon^{\mu\nu]}\right)n_{\mu\nu} = 0 \tag{49}$$

## V. APPLICATIONS

In this section we present some simple applications of this formalism.

### A. Michel monopole

We first discuss how the above theory behaves on the Michel monopole background, which is an exact solution

to conventional FFE [37]. This is a rotating magnetic monopole, and is a toy model for the magnetosphere outside of a rotating uniformly magnetized star. One should imagine that the core of the monopole is shielded by the star radius at $r = r_\star$.

In terms of the degrees of freedom $\epsilon$ and $\mu$, the solution can be written as:

$$\varepsilon = d(t-r)\wedge(dr - r^2\Omega\sin^2\theta d\phi) \qquad \mu = \frac{q}{r^2}. \tag{50}$$

which corresponds to

$$J = \frac{q}{r^2}d(t-r)\wedge(dr - r^2\Omega\sin^2\theta d\phi) \tag{51}$$

Though it is not necessary for our purposes, a valid choice of the foliation degrees of freedom is

$$\Phi_1 = \theta, \qquad \Phi_2 = \phi - \Omega(t-r), \qquad a = \frac{q}{r}dt \tag{52}$$

This is a solution to pure FFE. Does it remain a solution to the theory of free charges described above? As explained in [10], in the presence of the higher derivative coupling (21), a non-trivial $\mathbf{E}\cdot\mathbf{B}$ is created. This has no effect in the original FFE theory, but in the model with light charges it sources the $\Theta$ field, and we can no longer set it to zero. Instead we must solve the wave equation (49):

$$Q_0\nabla_\alpha\left[\nabla^{[\alpha}\frac{1}{|\mu|}\left(n^{\beta\sigma]}\Theta\right)\right]n_{\beta\sigma} = \frac{3}{\Lambda^2}\nabla_\sigma\left(\mu\nabla^{[\sigma}\epsilon^{\mu\nu]}\right)n_{\mu\nu} \tag{53}$$

where we have inserted the form of $Q(\mu)$ and $R(\mu)$.

In general, this is now a nonlinear problem that can only be solved numerically. In this work we tackle it by taking the scale $\Lambda^{-1}$ to be much smaller than any other scales in the problem, giving us a small parameter in which we can perform a perturbative expansion.

#### 1. Magnetic worldsheets and $dS_2$

We first examine the kinetic term appearing in (53). It is convenient to define outgoing time $u = t - r$ and a rescaled field

$$\Theta(t,r,\theta,\phi) \equiv \frac{1}{r^2}\tilde{\Theta}(r,t,\theta,\phi) \tag{54}$$

For future convenience, we define the wave operator on the left-hand side of (53) as $\square$:

$$Q_0\nabla_\alpha\left[\nabla^{[\alpha}\frac{1}{|\mu|}\left(n^{\beta\sigma]}\frac{\tilde{\Theta}}{r^2}\right)\right]n_{\beta\sigma} \equiv \square\tilde{\Theta} \tag{55}$$

We then find:

$$\square\tilde{\Theta} = \frac{2Q_0}{3q}\big[(1 - r^2\Omega^2\sin^2\theta)\partial_r^2 \\ - 2(\partial_u + \Omega\partial_\phi + r\Omega^2\sin^2\theta)\partial_r\big]\tilde{\Theta} \tag{56}$$

Note that there are no $\theta$ derivatives at all, and so each latitude completely decouples from the rest. However the rotation does introduce a $\phi$ derivative $\Omega\partial_\phi$. Indeed the full dependence on $(u, \phi)$ is now through the combination $(\partial_u + \Omega\partial_\phi)$

Curiously, the operator within square brackets is precisely the wave operator of two-dimensional de-Sitter space written in outgoing Eddington-Finkelstein coordinates:

$$ds^2 = -\left(1 - \frac{r^2}{L^2}\right) du^2 - 2dudr \qquad (57)$$

provided we make the substitution

$$(\partial_u)_{\text{dS}_2} \rightarrow (\partial_u + \Omega\partial_\phi)_{\text{Michel}} \qquad (58)$$

and where the de Sitter radius $L$ is

$$L = \frac{1}{\Omega \sin \theta} \qquad (59)$$

In particular, the location of the ($\theta$-dependent) de Sitter horizon is $r = (\Omega \sin \theta)^{-1}$, i.e. at the light cylinder. Heuristically, the light cylinder is the point where an observer rotating with the star would have to move faster than light. It acts as a horizon for Alfvenic perturbations of the background, and apparently also for the charge fluctuations developed in this paper.

The relevance of dS space to the Michel monopole was anticipated in [4], where it was pointed out that the intrinsic geometry of a Michel field sheet is indeed dS$_2$. It is thus unsurprising that charges that are confined to this world-sheet obey the de Sitter wave equation, albeit with an interestingly shifted notion of dS time (58). Studying the propagation of waves on this effective curved geometry will now require a treatment of boundary conditions at the de Sitter horizon. We would find it extremely satisfying if the physics of de Sitter horizons were to play an unexpected role in an understanding of the dynamics of pulsars, and hope to return to this point in the future.

Finally, as we will require it for the next section, we note that in the case that $\Theta$ depends only on $r$, the wave operator above reduces to

$$\Box\tilde{\Theta} \equiv \frac{2Q_0}{3q} \left[\left(1 - r^2\Omega^2 \sin^2 \theta\right) r^2\partial_r^2 - 2r\Omega^2 \sin^2 \theta\partial_r\right] \tilde{\Theta} \qquad (60)$$

The two independent solutions to this equation are

$$\Theta(r) = \frac{1}{r^2} \left(A + B \operatorname{arctanh}(r\Omega \sin \theta)\right) \qquad (61)$$

where we have expressed them in terms of the original field $\Theta$. Note that the term in $B$ is singular at the light cylinder.

### 2. Induced charges

We now return to our original problem of determining how the $\Theta$ field behaves in the presence of the Michel background. In particular, we will perform a perturbative expansion in powers of $\Lambda^{-2}$:

$$\Theta = \Theta_{(0)} + \Theta_{(2)}\Lambda^{-2} + \cdots \qquad J = J_{(0)} + J_{(2)}\Lambda^{-2} + \cdots \qquad (62)$$

Here $J_{(0)}$ is the force-free bare Michel solution, and it is consistent to set $\Theta_{(0)}$ to zero. To find the next order term, we insert this form into the wave equation (53). In terms of the wave operator (60), the equation for the term in $\Lambda^{-2}$ becomes

$$\Box\tilde{\Theta}_{(2)} = -\frac{8q\Omega\cos\theta}{r^3} \qquad (63)$$

To solve this equation, we need to specify boundary conditions on the field. We will demand that the solution remain regular at the light cylinder; this amounts to setting the coefficient $B$ to zero in (61).

At the neutron star surface $r = r_\star$, the physics is likely to be rather complicated and non-linear, and we do not at the moment have a good understanding of the correct boundary condition. For simplicity, we impose the Neumann boundary condition $\partial_r\Theta = 0$, noting that a Dirichlet condition would imply an explicit breaking of axial symmetry. Our result are largely insensitive to this choice, changing the final expression only by an $\mathcal{O}(1)$ factor.

With these choices, we find that to leading order in $\Lambda^{-2}$, the $\Theta$ field is:

$$\Theta(r) = \frac{3q^2(3r - 2r_\star)}{Q_0 r^2 r_\star \Lambda^2}\Omega\cos\theta + \cdots \qquad (64)$$

Using the identification (24), the electric current is

$$j_{\text{el}} = -18q^2(r - r_\star)\frac{\Omega\cos\theta}{r^4 Q_0 r_\star \Lambda^2}(\partial_t + \Omega\partial_\phi) \qquad (65)$$

This induced current is proportional to the co-rotating Killing vector of the Michel geometry $\partial_t + \Omega\partial_\phi$. It appears that the electric field created has populated space with this charge density, which subsequently attempts to rotate with the star. This rigid rotation happens faster and faster as we move out in $r$ until it reaches the speed of light at the light cylinder. The current vector is thus timelike at small $r$, null at the light cylinder, and spacelike outside it.

It would be very interesting to understand the consequences of this charge density in more physically realistic situations with less symmetry. This will likely require numerical simulation.

### B. Particle creation by wave scattering

FFE linearized about a homogenous magnetic field supports two types of linearly dispersing wave excitation; the transverse Alfven mode and the longitudinal magnetosonic (or fast) mode. The addition of the $\Theta$ field results

in a new gapless mode that disperses linearly along the field lines[5]. To leading order in the linearized theory, all of these modes are independent and do not mix. In this section we demonstrate that at nonlinear order, the scattering of two magnetosonic waves will generically result in particle creation.

Consider a homogenous background magnetic field of magnitude $B$ pointing in the $z$ direction; this corresponds to

$$\epsilon_0 = dt \wedge dz \qquad n_0 = dx \wedge dy \qquad \mu = B \qquad (66)$$

so that the magnetic flux is simply

$$J_0 = B dt \wedge dz \qquad (67)$$

We first review the dispersion relation of a single magnetosonic wave (see e.g. [11]). We work with the FFE equation of state $p(\mu) = \frac{1}{2}\mu^2$.

Let the spacetime dependence of the wave be $e^{ik_\mu x^\mu}$. We define the momentum parallel and perpendicular to the background field to be:

$$k_\parallel^\mu \equiv (h_0)^\mu_\nu k_\nu \qquad k_\perp^\mu = (h_0^\perp)^\mu_\nu k_\nu, \qquad (68)$$

where the projectors $h, h^\perp$ were defined in (16) and are here evaluated on the homogenous background field (66).

We now consider linearized perturbations around the background solution (66), $\epsilon \to \epsilon_0 + \delta\epsilon e^{ik_\mu x^\mu}$, $\mu \to B + \delta\mu e^{ik_\mu x^\mu}$. For convenience, let $a, b$ run over the directions parallel to the field $t, z$ and let $i, j$ run over $x, y$. By solving the linearized equations of motion, the components of $\delta\epsilon$ can be written in terms of the scalar perturbation $\delta\mu$:

$$\delta\epsilon_{ia} = -\frac{1}{k_\perp^2 B} k_i k^b \epsilon^0_{ba} \delta\mu \qquad (69)$$

(Both $\delta\epsilon_{ij}$ and $\delta\epsilon_{ab}$ identically vanish by the normalization and degeneracy constraints (15) on $\epsilon$ ). For the FFE equation of state all modes propagate at the speed of light, and the dispersion relation can be written in a manifestly Lorentz and rotationally invariant manner as

$$k_\parallel^2 = -k_\perp^2 \qquad (70)$$

Now we turn to the interactions with $\Theta$. A single linearized wave does not turn on the source term in (31); thus, as mentioned earlier, at the linearized level fluctuations of $\Theta$ and of the FFE modes decouple.

However if we consider the superposition of *two* magnetosonic modes with momenta $k_1$, $k_2$, then after some algebra we find for the source term

$$J^{\mu\nu} n_{\mu\nu}(x) = e^{i(k_1 + k_2)\cdot x} g(k_1, k_2)\delta\mu_1 \delta\mu_2 \qquad (71)$$

where the "interaction vertex" $g(k_1, k_2)$ is the following symmetric function of the momenta:

$$g(k_1, k_2) = -\frac{2}{\Lambda^2 B} \frac{\left(k_{\perp,1}^2 + k_{\perp,2}^2\right)}{k_{\perp,1}^2 k_{\perp,2}^2} n_0^{\mu\nu} k_{\mu,1} k_{\nu,2} \epsilon_0^{\mu\nu} k_{\mu,1} k_{\nu,2} \qquad (72)$$

Thus we see that non-trivial $\mathbf{E} \cdot \mathbf{B}$ has been created, and will generically create electric charges through the sourced wave equation (31). This is the main result of this section. It would be very interesting to understand if this provides a systematic way to understand an energy cascade from the magnetic field to electric charges, e.g. during FFE turbulence [39].

We briefly sketch how to go further and determine the $\Theta$ field itself. We must then solve the sourced wave equation (31), which to lowest order in the perturbations simply reduces to:

$$\frac{2Q_0}{3B} \left(\partial_t^2 - \partial_z^2\right) \Theta = -J^{\mu\nu} n_{\mu\nu} \qquad (73)$$

As usual this can be solved by introducing a Green's function $G_\Theta(x, y)$ for $\Theta$, which is a delta function in the transverse directions but propagates gaplessly along the field lines:

$$G_\Theta(x, y) \equiv \frac{3B}{2Q_0} \delta^{(2)}(x^i - y^i) \int \frac{d^2 p}{(2\pi)^2} e^{ih_0^{ab} p_a (x-y)^b} \frac{1}{h_0^{ab} p_a p_b} \qquad (74)$$

A particular solution to (73) is then simply

$$\Theta(x) = -\int d^4 y\, G_\Theta(x, y) J^{\mu\nu} n_{\mu\nu}(y) \qquad (75)$$

Combining this expression with (71), it should be clear that we are constructing a classical solution by evaluating tree-level Feynman diagrams; in particular, we have shown that there is an amplitude for two magnetosonic waves to scatter and create particle charge, and thus the Feynman rules for this theory have a vertex of the form shown in Figure 1.

We note that this is a sort of "inverse pion decay": conventionally, the $\pi^0$ (which, like $\Theta$, is roughly a Goldstone mode for a spontaneously broken axial symmetry) decays to two electromagnetic excitations (photons) through a channel mediated by the axial anomaly (34). Here two electromagnetic excitations (magnetosonic modes) combine through the anomaly-mediated channel (33) to create an excitation of $\Theta$.

It would be very interesting to further develop and understand the physical consequences – if any – of this diagrammatic expansion.

---

[5] We can compare this with the results of [38], which studies a model of spontaneously broken axial symmetry coupled to free Maxwell EM with a background magnetic field. They find a mixing between an unscreened $E_\parallel$ and the axial Goldstone, resulting in a quadratically dispersing mode. Our results differ because our electromagnetic sector is governed by FFE, where $E_\parallel$ is already gapped out and cannot mix with anything.

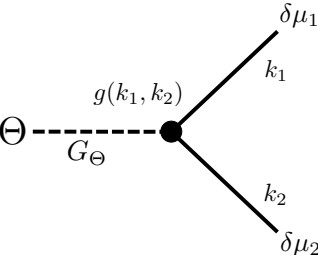

FIG. 1. "Feynman" diagram associated with the evaluation of the term computed in (75); solid lines indicate magnetosonic modes (where in our kinematics the propagators have been amputated) and dotted line indicates $\Theta$ propagator $G_\Theta$.

## VI. CONCLUSIONS

In this work, we showed how to couple extra light electric charges – charges that are not in equilibrium with the plasma – to the FFE EFT of [10]. We conclude with some directions for future work.

One original motivation for this study was astrophysical. Ideally, one might hope that this model could provide a useful caricature of the dynamics of compact astrophysical objects, where open questions remain regarding (e.g.) the origins of coherent radiation and the production of particle winds [7–9].

In particular, the truncated model described in Section IV provides a concrete deformation away from FFE, parametrized by a single scale $\Lambda$ that controls the scale at which unscreened electric fields can self-consistently appear. Such unscreened fields will subsequently pair-create and accelerate charges, as captured by ripples in the collective field $\Theta$. It would be very interesting to understand the physical consequences of such nonlinear dynamics in more realistic geometries with less symmetry than the Michel monopole. This will likely require numerical simulation. We note that the viewpoint taken here – deforming in a controlled manner *away* from FFE – is the opposite to that taken in usual particle-in-cell simulations (see e.g. [40, 41]), where one starts microscopically with free Maxwell electrodynamics coupled to charge dynamics and *arrives* (at long distances) at FFE.

One clear deficiency of the system is the fact that it provides a clean description only in situations when the mass of the electron can be neglected. It is somewhat nontrivial to study the electron mass as a perturbation; it seems that any attempt to do so and thus break the axial shift symmetry (25) also ultimately correlates fluctuations on different field lines, essentially because a massive electron has a finite transverse radius.

For any potential application to real-life systems, it is also clearly crucial to understand the magnitude of scales such as $\Lambda$. Such higher-derivative corrections can in principle be precisely matched to a UV description (e.g. QED) using the analogue of hydrodynamic Kubo formulas (see e.g. [42] for a review). We hope to report on this in the future.

## ACKNOWLEDGEMENTS

I am very thankful to S. Gralla for collaboration on the initial stages of this project and helpful conversations throughout. I am also grateful for conversations on related issues with S. Grozdanov, U. Gursoy, D. Hofman, A. Lupsasca and N. Poovuttikul. I thank the Aspen Center for Physics (which is supported by National Science Foundation grant PHY-1607611) where part of this work was performed in the working group "Applying tools from high-energy theory to problems in astrophysics", and where my trip was supported by a grant from the Simons Foundation and an International Engagement Grant from Durham University. I am supported in part by the STFC under consolidated grant ST/L000407/1.

## Appendix A: Comparison to microscopic strong-field bosonization

Here we compare our theory with the top-down construction of [17], which follows earlier foundational work in [15] (see also [16] for a similar computation in a different context).

We briefly review [17]: there QED with electromagnetic coupling $e$ is studied with a single species of massive Dirac fermion on a constant magnetic field background. The fermion is decomposed into Landau levels, and then each fermion state in the lowest Landau level is bosonized into a massless scalar $\phi_i$ with a kinetic term with derivatives only along the magnetic field directions, where $i$ runs over the $N = \frac{eBA}{2\pi}$ states in the lowest Landau level. It is then assumed that all of the bosons $\phi_i$ move in sync, i.e.

$$\phi_1 = \phi_2 = \cdots = \phi_N = \Phi \ . \tag{A1}$$

An effective action is constructed for $\Phi$ and its coupling to the electromagnetic field, and it is then assumed that the fields are allowed to vary slowly in the transverse directions to obtain a 4d dynamical theory. Derivatives of $\Phi$ in the transverse directions do not appear in this model.

This sounds structurally similar to our EFT. We now attempt a comparison: this is possible only for small fluctuations around a homogenous background (66). $\Phi$ results in an effective electric current of the form

$$j_{\text{el}}^\mu = -\frac{e^2}{8\pi^2}\epsilon^{\mu\nu\rho\sigma}\partial_\nu\Phi F_{\rho\sigma} \tag{A2}$$

Comparing this to our (24), we see that these take the

same form if we identify

$$\Theta = \frac{e^2}{4\pi^2}\mu\Phi \qquad (A3)$$

(Here $\mu$ is taken to be constant). Comparing now our kinetic term (29) with that in Eq (3.2) of [17], we see that they agree if

$$Q(\mu) = \frac{2\pi^2}{e^3}\frac{1}{|\mu|}, \qquad (A4)$$

thus motivating the value of $Q_0$ (41) described in the main text.

We now discuss the issues with such a comparison. Firstly as our $\Theta$ is gapless, the fermion mass $m$ in [17] must be ignored; as mentioned in that work, this appears to be a necessary condition for a force-free limit in any case. A further issue is that even if $m$ is set to zero, linearized fluctuations of $\Phi$ coupled with the electromagnetic field are still gapped, for essentially the same reason that the Schwinger model shown in (2) is gapped. This is clearly not the case for our gapless $\Theta$ field, and we believe that this occurs because the dynamics of the $N-1$ fields $\phi_i$ (which remain gapless [14]) have been neglected.

We find it plausible that the gapped mode in [17] is eaten by the electromagnetic field and helps to sustain the FFE architecture in the manner originally proposed in [15], whereas the remaining $N-1$ modes remain gapless and assemble eventually into our $\Theta$ field, carrying axial current. It would be very interesting to further verify this picture.

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
