# Peer review of "Effective description of non-equilibrium currents in cold magnetized plasma"

_SciPost Physics_

## Round 1 · Referee Report · Anonymous · 2021-4-30

Strengths

1- Makes progress in an important astrophysical question (What is the non-linear dynamics of magnetic field lines in a plasma in the presence of out-of-equilibrium charges?) using a clean theoretical approach.
2- The formal development of the effective field theory is accompanied by several applications.

Weaknesses

1- The problem of extending the framework to massive charge carriers is left unresolved.
2- While the paper contains quite a bit of background, the discussion of the EFT of force free electrodynamics is not self-contained (as the author also points out).

Report

This paper determines the dynamics of non-equilibrium currents in cold magnetized plasma using an effective field theory (EFT) description. It builds on prior work of the author that developed an EFT description for force free electrodynamics, and solves the (astro)physically important problem of how to couple this EFT to non-equilibrium currents. Both the original EFT and its coupling to non-equilibrium currents is governed by unconventional and intricate symmetries that the author presents very clearly. The paper would deserve publication even as purely theoretical exercise, and its value is further enhanced by the promise of applying its formalism to astrophysical problems and its preliminary discussion of some applications.

Requested changes

1- Present the EFT of force free electrodynamics in a more self-contained manner, especially the meaning of the EFT degrees of freedom (perhaps through a figure).
2- Explain explicitly (around eqs. (18)-(22)) what the EFT is an expansion in, i.e. what physical scale makes the derivative expansion dimensionless. (I understand that this issue is discussed later in the paper.)
3- Explain why in Sec IV. one higher derivative term is kept. Why isn't it enough to analyze the coupled theory of FFE and $\Theta$?

---

## Round 1 · Referee Report · Anonymous · 2021-6-2

Strengths

1) The paper presents clearly the background and the assumptions made in constructing the new model.
2) The model addresses an important open problem and has potential phenomenological applications

Weaknesses

1) The main weakness, as the author himself recognizes and discusses in section III.E, is the rather ad-hoc way in which the scalar sector is introduced, without fitting into the framework of derivation from symmetry principles. This is fine for a phenomenological model but leaves open the question of universality of the construction.
2) It is not very clear, or clearly explained, what is the regime of validity of the model. Since the scalar sector describes out-of-equilibrium currents, one is not describing fluctuations around equilibrium as in conventional hydrodynamics, so it is not obvious what is the relevant expansion parameter.

Report

The paper presents a development of the theory of force-free electrodynamics, seen from the modern point of view of generalysed symmetries. Specifically, the author proposes an effective action
that couples the FFE sector (which describes a magnetically-dominated cold plasma) to a scalar field, which is assumed to be a collective field encoding the dymamics of gapless modes coming from the zero-Landau levels of electrons in the magnetic field. Technically, the main assumption to constrain the effective action is the scalar-shift symmetry postulated in eq. (25).
The author then shows that the resulting physics, to the lowest order in derivatives, reproduces qualitatively the physics expected from the axial anomaly of the fermions localized on the field lines of the magnetic field. Some physical consequences of the formalism are worked out in two examples.
The model is supposed to have real-world applications in astrophysical systems, in situations where FFE is not sufficient to account for the phenomenology.
The paper thus represents a significant step towards a more complete description of such systems.

Requested changes

1) The author should address the issue of the regime of validity of the model, and clarify what is the relevant expansion parameter.
2) It should also be better explained under what conditions the charged matter can be included in the way presented in the model, but still without spoiling the assumptions of force-free electrodynamics.
3) There are several terms in the effective action, that are all supposed to be at leading order in derivatives, but have different number of derivatives, can the author explain how the counting works?
4) Can the author elaborate more on why the dependence on the fields $\Phi_{1,2}$ in the action has to come only through the normal $n_{\mu\nu}$?
5) The function $R(\mu)$ has an expansion as in eq. (40), however couldn't there be also a constant, $\mu$-independent term?

---

## Editorial Decision

resubmitted